# Manufacturing and Characterization of Modified Wood with In Situ Polymerization and Cross-Linking of Water-Soluble Monomers on Wood Cell Walls

**DOI:** 10.3390/polym14163299

**Published:** 2022-08-12

**Authors:** Jihang Hu, Zongying Fu, Xiaoqing Wang, Yubo Chai

**Affiliations:** Research Institute of Wood Industry, Chinese Academy of Forestry, Beijing 100091, China

**Keywords:** poplar, cell walls, cross-linking modification, wood properties

## Abstract

Fast-growing plantation wood has poor dimensional stability and easily cracks, which limits its application. As wood modification can improve the dimensional stability, strength, and other properties of wood, it has been extensively used. In this study, 2-Hydroxyethyl methacrylate (HEMA) and glyoxal were applied to treat poplar wood (*Populus euramevicana cv.I-214*) by using vacuum pressure impregnation to improve its dimensional stability. The weight percentage gain (WPG), anti-swelling efficiency (ASE), water absorption rate (WAR), leachability (L), and other properties of modified wood were examined. Results showed that the modifier was diffused into the cell walls and intercellular space and reacted with the wood cell wall after heating to form a stable reticular structure polymer which effectively decreased the hydroxyl content in the wood and blocked the water movement channel; thus, further improving the physical performance of wood. These results were confirmed by scanning electron microscopy (SEM), energy-dispersive X-ray spectroscopy (EDX), X-ray diffraction (XRD), Fourier-transform infrared spectroscopy (FTIR), and nuclear magnetic resonance (NMR). When the ratio of the modifier was 80:20, the concentration of the modifier was 40%, and the curing temperature was 120 °C, the modified poplar had the best performance, which showed a low WAR (at its lowest 58.39%), a low L (at its lowest 10.44%), and a high ASE (of up to 77.94%).

## 1. Introduction

Wood, as a significant natural and renewable material, has been extensively used in furniture, decoration, construction, and other fields because of its outstanding high strength-to-weight ratio, easy processing, beautiful appearance, and affordability [1,2]. Owing to the capability of carbon dioxide storage, wood and wood utilization are more friendly to the environment [3]. Fast-growing wood is considered an alternative source of natural wood due to its short planting time and is an ideal material for a sustainable society [4]. Nevertheless, the application of fast-growing wood is restricted by its poor dimensional stability, poor physical and mechanical properties, and easy cracking. However, wood modification technology could significantly improve the performance and quality of fast-growing wood, which is of great significance in broadening its application range and extending its service life [5]. Therefore, using modified fast-growing wood can relieve the shortage of wood supply and protect the over-exploitation of natural resources as an alternative means. Among numerous wood modification methods, chemical cross-linking modification has received more attention, with the advantages of a simple process, significant modification effect, and excellent stability [6,7]. It has been found that the strength and density of wood increase if the wood modifier is able to penetrate and enlarge the wood cell wall. If the modifier can react with the cell wall constituents, the modification impact is substantially improved, and the stability is extended [7,8,9]. The dimensional stability of wood can be enhanced by reducing the hydrophilic -OH group of cell wall components or bulking the cell wall with a modifier. Thus, it is a significant issue for the wood industry to ensure that the wood is non-toxic and environmentally sustainable during the treatment and use processes, and that it will not cause damage to the environment and the human body.

Hydroxyethyl methacrylate (HEMA) includes numerous active hydroxyl groups, which contribute to its biocompatibility. Additionally, since it is simple for the cross-linking agent to self-polymerize or create a network structure, it has excellent mechanical qualities. Due to the great strength and biocompatibility of Poly (2-hydroxyethyl methacrylate) (pHEMA) hydrogels, they have been used in a variety of biomedical applications [10,11,12]. Additionally, there have been several uses for wood modification in recent years [13]. Due to its high concentration of active hydroxyl groups, HEMA was selected as the main modification to increase the performance of wood cell walls [14]. It is expected that the hydroxyl groups will enhance the polymers’ hydrogen bonding capabilities with wood components [15,16]. Glyoxal (GA) is a widely used cross-linking agent for the preparation of high-molecular-weight gel materials. It has excellent graft performance and biocompatibility. Due to its excellent cross-linking performance and low toxicity and volatility, it is widely used in the textile, paper, and chemical industries, etc. [17,18]. Modifying wood to enhance the dimensional stability and water resistance of wood using a cross-linking agent is an efficient modification strategy. As a result, treating wood with glyoxal may significantly enhance its physical and mechanical qualities [19]. Glyoxal has two interconnected aldehyde groups in its molecular structure and has high chemical activity and all the chemical properties of aldehydes. It is widely used in the wood industry to replace formaldehyde in the synthesis of resins [20,21,22]. Glyoxal is a well-studied bi-functional aldehyde which has the competent ability to react with hydroxyl groups of wood cell wall polymers and enhance dimensional stability, water repellence, and durability of wood [19]. However, these improvements are often accompanied by brittleness and strength losses in the wood [23].

The impregnation of wood cell walls with hydroxyethyl methacrylate and glyoxal resulted in the modification of the cell walls, and thermally induced in situ polymerization resulted in the formation of modified materials with outstanding characteristics. The purpose of this work is to enhance the quality of wood (dimensional stability and water absorption) by grafting polymers onto the cell walls of wood, expanding the cell walls, and converting hydrophilic hydroxyl groups into bigger hydrophobic groups. This work advances fast-growing-wood modification by introducing a novel research strategy. Modified wood is projected to be employed in furniture manufacture and interior decorating, broadening the spectrum of fast-growing wood’s applications and enabling the replacement of natural forest wood with fast-growing wood.

## 2. Materials and Methods

### 2.1. Materials

The poplar wood (*Populus eurame**ricana cv.I-214*) was collected from the Sun jia zhuang forest farm in Yi County, Hebei province, China. The average tree age and diameter at breast height are about 20 years and 25~33 cm, respectively. Additionally, the air-dry density is about 0.35 g/cm^3^. Mature sapwood was randomly picked and sawed into 20 mm × 20 mm × 20 mm pieces, which needed to be completely dry before the impregnation experiment. All chemical reagents were analytical grade. HEMA (Kangboshunda Chemical Products Co., Ltd., Beijing, China), glyoxal of chemical grade (40%), analytical-grade calcium chloride (CaCl_2_), hydrogen peroxide (30% aqueous solution, H_2_O_2_), and ammonium persulfate ((NH_4_)_2_S_2_O_8_) (Aladdin Co., Ltd., Shanghai, China; Beijing, China).

### 2.2. Wood Modification Methods

Modifiers included HEMA and glyoxal, and the compound initiator consisted of CaCl_2_, H_2_O_2_, and (NH_4_)_2_S_2_O_8._ The modifiers were prepared with varied HEMA to glyoxal ratios (HEMA:GA = 100:0, 90:10, 80:20, 70:30, 60:40) and different concentrations (10%, 20%, 30%, 40%). The amount of compound initiator was 3% of the mass of the modifier, and the mass ratio of CaCl_2_, H_2_O_2_, and (NH_4_)_2_S_2_O_8_ is 1:1:1. CaCl_2_, H_2_O_2_, and (NH_4_)_2_S_2_O_8._ One parameter was taken as the variable and the others were left unchanged. Specific parameters have been shown in Table 1.

Wood samples were impregnated respectively with the above prepared solution, through the vacuum-pressure method as vacuuming (0.1 MPa, 1 h) → liquid injection → pressurization (1.0 MPa, 8 h) → pressure relief. After releasing the pressure, the wood sample was air-dried for 5 days and then heated to 40 °C for 24 h to initiate the reaction between the modifier and the wood cell wall. They were then heated for 6 h at 80 °C, 100 °C, 120 °C, and 140 °C, respectively, to complete the polymerization process (Figure 1). Finally, all the samples were dried at 103 ± 2 °C to achieve the same quality, except for the samples in 120 °C and 140 °C conditions. Each group had 10 duplicate samples, and had its statistical average value taken.

### 2.3. Characterization

Scanning electron microscope (SEM) and Energy-dispersive X-ray spectroscopy (EDX). SEM (Hitachi SU8020, Toyko, Japan) was used to investigate the distribution of modifiers in wood cell walls at an accelerating voltage of 10 kV. Samples of tangential modified and unmodified wood (5 mm × 5 mm × 5 mm) were coated with a platinum layer on an aluminum SEM stub. Additionally, EDX coupled with SEM was utilized to detect the concentration of the characteristic elements of wood samples.

X-ray diffraction (XRD) analysis was performed by a Bruker D8 Advance diffractogram from Germany. Cu-Kα radiation with graphite monochromator, 40 kV, 40 mA, and 2θ scan range of 5–45° (2θ) with a scanning speed of 2°/min were preset parameters. On the scanning curve, there was a maximum peak of (002) diffraction near 2θ = 22° and a minimum near 2θ = 18°. The empirical formula proposed by Segal et al. [24] was used to calculate the relative crystallinity. The crystallinity index (CrI) was calculated as follows:(1)$$\mathrm{CrI}\;(\%)=\frac{100\left(I_{002}-I_{am}\right)}{I_{002}}$$
where *I*_002_ is the intensity of the diffraction angle of (002) crystal plane (2θ = 22°), and *I**_am_* is the scattering intensity of the amorphous background diffraction degrees (2θ = around 18°). The microfilament angle of the sample was calculated according to the Cave 0.6T method [24].

Fourier-Transform Infrared Spectroscopy (FTIR) was used to characterize the chemical composition of the modified wood samples. All samples (three samples each for modified and unmodified wood) were crushed to 100 mesh and ground using KBr pellets. A FTIR analysis was performed using a FTIR Spectrometer (Nicolet Nexus 6700, Madison, WI, USA) in the range of 400 to 4000 cm^−1^ in diffuse reflection mode with a 4 cm^−1^ resolution and 64 scans.

Nuclear magnetic resonance spectroscopy (NMR). ^13^C NMR spectra of the modified wood samples were recorded on a Bruker Avance AVIII 400 MHz NMR spectrometer. The modified and unmodified poplar wood flour with 160 mesh was prepared for an NMR. The solid-state ^13^C magic angle spinning (MAS) NMR spectra of the modified wood samples were examined and compared with unmodified wood samples. The MAS speed was set at 5000 Hz and scanned 8318 times at room temperature.

### 2.4. Wood Performance Evaluation

The samples with a size of 20 mm × 20 mm × 20 mm (longitudinal × radial × tangential) were prepared for weight percentage gain (WPG), bulking effect (BE), anti-swelling efficiency (ASE), water absorption (WAR), and leachability (L) tests. Ten replicates were conducted for each group, and the statistical average value was taken.

The weight percentage gain (WPG) was calculated according to the absolute dry mass before and after modification, as seen below (Formula (2)).
(2)$$\mathrm{WPG}\;(\%)=\frac{100\left(W_{1}-W_{0}\right)}{W_{0}}$$
where *W*_0_ and *W*_1_ are the weight of oven-dried wood samples before and after treatment, respectively.

The bulking effect (BE) test is used to characterize the size changes of wood before and after modification. The BE was calculated (see Formula (3)).
(3)$$\mathrm{BE}\;(\%)=\frac{100\left(V_{1}-V_{0}\right)}{V_{0}}$$
where *V*_0_ and *V*_1_ are the volume of oven-dried wood samples before and after treatment, respectively.

By completely immersing the modified and unmodified samples for 120 h, the volumes of the modified and unmodified samples before and after immersion were measured. From these volumes, ASE can be calculated as follows:(4)$$VSE\;(\%)=\frac{100\left(V_{a}-V_{b}\right)}{V_{b}}$$
where *V_a_* and *V_b_* are the volumes of wood samples after and before immersion, respectively.
(5)$$\mathrm{ASE}\;(\%)=\frac{100(VSE_{u}-VSE_{t})}{VSE_{u}}$$
where *VSE_u_* and *VSE_t_* are the volumetric swelling efficiencies of untreated and treated wood samples, respectively.

The water absorption rate was calculated according to the “Determination method of water absorption of wood” (GB/T 1934.1-2009). First, the modified samples were numbered and weighed. The specimens were placed at least 50 mm below the surface of the water and soaked for 10 days, and then the samples were removed from the water, and the surface was wiped with a tissue and weighed again. The WAR was calculated as:(6)$$\mathrm{WAR}\;(\%)=\frac{100\left(W_{2}-W_{1}\right)}{W_{1}}$$
where *W*_1_ and *W*_2_ are the absolute dry mass after modified and the mass after soaking of the modified wood specimens, respectively.

After calculating the WAR, all samples were oven-dried at 103 ± 2 °C to a constant weight. The leaching rate due to water immersion was calculated as follows:(7)$$L\;(\%)=\frac{100\left(W_{1}-W_{3}\right)}{\left(W_{1}-W_{0}\right)}$$
where *W*_3_ is the absolute dry weight of a modified wood sample after a 10 d water immersion.

## 3. Results and Discussion

### 3.1. Distribution of Modifiers

Cross-sectional microphotographs of both treated and untreated wood were obtained by SEM and EDX. As illustrated in Figure 2a,b, the cell walls in modified wood were thicker than that of unmodified wood, and unmodified wood had distinct gaps between the cell walls that were not present in modified wood. The untreated cell walls had a mean thickness of 5.04 ± 0.96 μm, and the treated cell walls had a mean thickness of 7.74 ± 0.57 μm (Table 2). The resulting polymer from the copolymerization of HEMA and glyoxal was grafted onto the wood cell wall, thereby improving the interfacial compatibility between the polymer and the wood cell wall without obvious gaps. According to the energy spectrum analysis (Figure 2e,f), the carbon, oxygen, and nitrogen contents of the modified wood cell walls were greater than those of the unmodified wood. This result indicates that the modifier was predominantly distributed in the cell walls and, to a lesser extent, in the cell lumen.

### 3.2. XRD Analysis

As seen in Table 3 and Figure 3b, the peak width of the X-ray diffraction pattern of poplar increased from 1.74 to 2.49, and the microfibril angle increased from 21.86° to 29.70°. In this hypothesis, it is believed that when the modifier causes the cell wall to swell, some of the amorphous region components are dissolved, resulting in the original arrangement of the microfilaments changing.

The XRD spectrum (Figure 3a) showed that the maximum diffraction values of the crystalline region of the modified wood (002) were around 2θ = 22°, which implies that the treatment had no effect on the crystalline region, and thus there was no change in the distance between the crystal layers. In terms of diffraction intensity, the treated wood increased a little, indicating a slight effect on the amorphous region of the wood microfibrils. Troughs appeared near 2θ = 18°, and the difference from untreated wood was less than 9%. Therefore, from the perspective of a crystal structure, there is no difference in the crystal structure of the cellulose of treated wood, and it is still cellulose I-type structure [25].

The relative crystallinity of wood is a critical parameter for describing the supramolecular structure of cellulose, which is intimately related to its physical and chemical properties. Additionally, relative crystallinity is indispensable for comprehending wood alteration, processing, and utilization [26]. The relative crystallinity of modified wood increased from 43.15% to 54.77% (Table 3) because the modifiers entered the cell wall of the wood and affected the crystalline structure of the cellulose molecular chain. When the amorphous fiber recombination occurred after the temperature curing, a stronger bond was established. As the relative crystallinity improved, the dimensional stability of the wood also improved, leading to further improvements in its physical properties. With the increase in wood crystallinity, the dimensional stability and nuclear density of wood also increased, which is consistent with the results of the physical properties evaluation of modified wood.

Despite the slight change, high-temperature curing alters the width of the crystallization region in wood. The half-peak width of the diffraction peak of the (002) crystal plane was used to calculate the width of the crystalline region of the wood [27]. When the reaction temperature reached a certain level, the crystalline zone and amorphous zone of the fibers changed, and the width of the crystalline zone of the modified materials increased. This was consistent with the findings of the microfibril angle and relative crystallinity investigation.

### 3.3. FTIR Analysis

The FTIR spectra in Figure 4 clearly show that the absorption peaks near 3373 cm^−1^ became weaker after grafting with HEMA and glyoxal, and the broad bands at 3200–3600 cm^−1^ were attributed to primary amine O-H bands, which indicated that the -OH on the wood was involved in the grafting reaction [28,29].The strong absorption peaks of C=C and C=O appeared at 1666 cm^−1^, and 1627 cm^−1^, mainly caused by the stretching vibration of the carbonyl and aldehyde groups in HEMA and glyoxal. As shown in Figure 4, the peak at 1594 cm^−1^ corresponds to −C=O stretching and disappears from the spectrum of the modified material. This is due to the oxidation reaction between the modifier and the wood. In addition, the 898 cm^−1^ absorption peak in the modified wood was weakened, indicating that the modifier had a certain influence on the cellulose structure of the wood [30,31]. FTIR spectroscopy supported the XRD results.

Both HEMA and glyoxal are highly reactive, and this functional group can react with hydroxyl groups in wood. Under high temperature and catalyst action, the hydroxyl groups between the wood modifiers can be cross-linked to form a three-dimensional network polymer under heating conditions. It is because of the chemical cross-linking reaction of the reactive modifiers in the wood and the uniform distribution in the wood that the physical and chemical properties of wood were comprehensively improved.

### 3.4. ^13^C NMR Analysis

In order to further study whether wood reacts with HEMA/GA, and confirm the conclusion of FTIR, NMR (Figure 5) was used to study whether there is a new chemical shift in the modified wood and whether there is a chemical reaction in the modified wood.

The absorption peaks at 88 and 83 ppm were enhanced, indicating that the chemical cross-linking reaction occurred between the modifier and C4 in crystalline and non-crystalline areas, which rearranged the molecules in the non-crystalline area and formed a new crystalline area, thus increasing the crystallinity of the wood. This result is the same as the XRD result. The characteristic peaks at 55 ppm and 120–170 ppm have noticeable chemical shifts compared with the unmodified materials, and here represent the methoxy carbon and aromatic carbon components of lignin, indicating that a cross-linking reaction between the modifier and the carbon in lignin has occurred [32,33,34].

### 3.5. Effect of the Ratio of HEMA and Glyoxal on Properties of Modified Wood

Due to the porous nature of wood and the higher concentration of free hydroxyl groups, it is excellent at storing water molecules and has high hygroscopicity. When wood absorbs water, water molecules combine with free hydroxyl groups, causing hemicellulose and cellulose to swell, their intermolecular hydrogen bonds to break, and more hydrophilic active sites to be exposed, thereby increasing water absorption and swelling of wood and resulting in deformation [35]. The main objectives of wood modification in this study are to reduce the number of hydroxyl groups and channels of water movement.

Figure 6 reflects that the ratio of HEMA and glyoxal has an influence on wood bulking effects BE, ASE, WPG, WAR, and L, while other factors remain unchanged (the modifier concentration was 20%; the curing temperature was 120 °C).

WPG and BE increased with the addition of glyoxal, demonstrating that glyoxal promotes the modifier into the wood cell wall. The BE reached its maximum when the HEMA to glyoxal ratio was 80:20, which indicates that the addition of glyoxal induces more modifiers into the wood cell wall and that the swelling of the cell wall by the modifier has reached its limit. Although WPG increased with increasing glyoxal (HEMA:GA = 70:30), BE decreased slightly, indicating that the modifier entered the cell cavity. When the ratio was 60:40, WPG decreased slightly, which could be attributed to an overabundance of glyoxal, which caused self-polymerization of glyoxal, thus preventing the modifier from entering the wood. When the HEMA to glyoxal ratio was 70:30, the variation trend was consistent with that of WPG, ASE, and WA, where the best values were 31.88%, 74.18%, and 107.44%, respectively. This is because modifiers can penetrate wood cell walls and react with the cell wall components of the wood to improve cell wall performance. Glyoxal is a reactive cross-linking agent for bifunctional groups, while HEMA is a cross-linking agent for bifunctional groups. By formulating polymers with the two ingredients, HEMA and glyoxal, and using hydroxyl cross-linking reactions, vinyl and hydroxyl groups can polymerize to form a three-dimensional structure that helps improve the cohesion strength of polymers, thus increasing the physical properties of wood [36]. Another approach is for the modifier to polymerize and block the transport channels of water in the wood; therefore, to improve the dimensional stability of the wood, these pores include vessels, pits, etc. Because of this, the water absorption rate of wood was reduced, which ultimately benefited the product’s overall shelf life.

Furthermore, both HEMA and glyoxal can react with hydrophilic hydroxyl groups on wood, thus reducing the ability of wood to absorb water. ASE increased to 74.18% when the ratio was 70:30. Compared to untreated wood (168.60%), the water absorption of WA (107.44%) decreased by 36.28%. Basically, the changing trend in leachability was the same as BE. The leachability was best when the ratio of 80 to 20 showed the modifier had the best retention ratio on the cell wall. A comprehensive evaluation of the physical properties of ASE, WA, and L requires an optimal modifier ratio of 70:30.

### 3.6. Effect of Modifier Concentration on Properties of Modified Wood

Figure 7 reflects that the modifier concentration influences the bulking effect, dimensional stability, weight gain, water absorption rates, and leachability of wood, while other factors remained unchanged (the ratio was 80 to 20; the curing temperature was 120 °C).

As the modifier concentration increased, WPG and BE both exhibited a positive correlation trend. The results indicated that the higher the concentration, the more modifier was infused into the wood and the greater the swelling of the cell wall. In Figure 7, an increase in modifier concentration results in a higher WPG. When the modifier concentration was 10%, the WPG was 14.22%, while ASE was 56.59%, indicating the high efficiency of HEMA/GA in improving the dimensional stability of poplar wood. When the concentration reached 40%, ASE and WPG increased to 77.94% and 60.19%. In addition, the cross-linking degree increased with increasing modifier concentration, and the oligomer decreased, reducing the modifier loss rate from 33.10% (the modifier concentration was 10%) to 10.38% (the modifier concentration was 40%). The initiator successfully grafted the modifier onto the hydroxyl groups in wood, thereby reducing the number of hydroxyl groups and the hygroscopic rate of the wood. Obviously, when the concentration was changed, the water absorption of wood decreased by 65.4% compared to untreated wood. The modifier was fixed into the wood, blocking water flow, causing a decrease in the water absorption of the wood. After a comprehensive inspection of ASE, WA, and L, it was determined that the best performance was obtained at a modifier concentration of 40%. The ASE obtained in this work was much higher than in other studies [7,17,37,38].

### 3.7. Effects of Curing Temperature on Wood Properties

The effects of different curing temperatures on the properties of modified poplar wood were investigated. Other technological parameters, including HEMA to glyoxal ratio (80:20) and modifier concentration (20%), remained unchanged. The modification results of wood are impacted by temperature, which is directly related to the activity of the catalyst and the degree of cross-linking of the modifier. As can be seen from Figure 8, the curing temperature has no significant effect on BE. With a temperature exceeding 120 °C, WPG declined slightly due to the rapid volatilization of glyoxal and the degradation of wood cell wall components. The reactivity of HEMA and glyoxal, as well as the cross-linking with the wood composition, increased as the temperature increased. To ensure that the synthetic property of wood was the best, the curing temperature was set at 120 °C. At a curing temperature of 120 °C, the ASE of wood can reach 68.50%, with a leachability of 19.33%.

## 4. Conclusions

In this study, the dimensional stability of wood was improved by the in situ polymerization of water-soluble monomers in water. A water-soluble monomer and cross-linker were injected into the wood cell walls and activated cross-linking reactions to form interpenetrating polymer network structures. The polymer network blocked the partial pore and reduced wood hydroxyl, which simultaneously and significantly increased the wood’s transverse connection, dimension, and stability. The results showed that the best performance of ASE, WAR, and L was achieved at 77.94%, 58.39%, and 10.44%, with the experimental parameters of a 70:30 modifier ratio, 40% modifier concentration, and 120 °C curing temperature, respectively. Compared to unmodified wood, modified wood has thicker cell walls and smaller cell gaps. In addition, crystallinity, crystalline zone width, and the microfiber angle were all greater in modified wood, indicating that the modifiers are efficiently dispersed into the cell walls. The FTIR analysis reveals that the -OH of the modified wood, which is involved in the reaction of HEMA and glyoxal, is significantly reduced. Furthermore, the characteristic chemical bonds C=C and C=O in the modifiers were significantly increased, indicating that the modifiers strongly bonded to the wood. The results from FTIR and NMR analyses indicate that the modifiers existed in the wood cell wall and reacted with wood components. The enhanced absorption peaks at 88 and 83 ppm indicate the formation of a new crystallization zone, and consequently, the crystallinity of the modified material increased. Compared with the unmodified wood, the characteristic peaks of 55 ppm and 120–170 ppm showed significant chemical shifts, indicating a cross-linking reaction between the modifier and the carbon in the lignin. The results of the NMR analysis verified the results of FTIR and XRD.

## Figures and Tables

**Figure 1 polymers-14-03299-f001:**
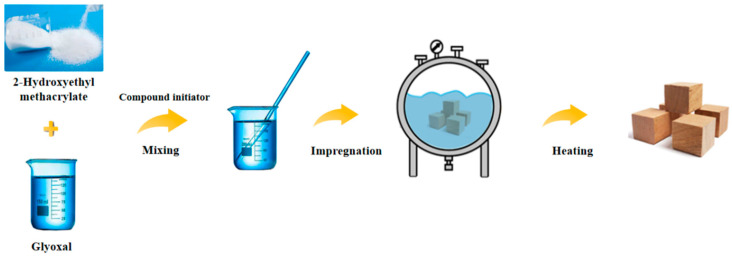
Schematic diagram of the wood modification process.

**Figure 2 polymers-14-03299-f002:**
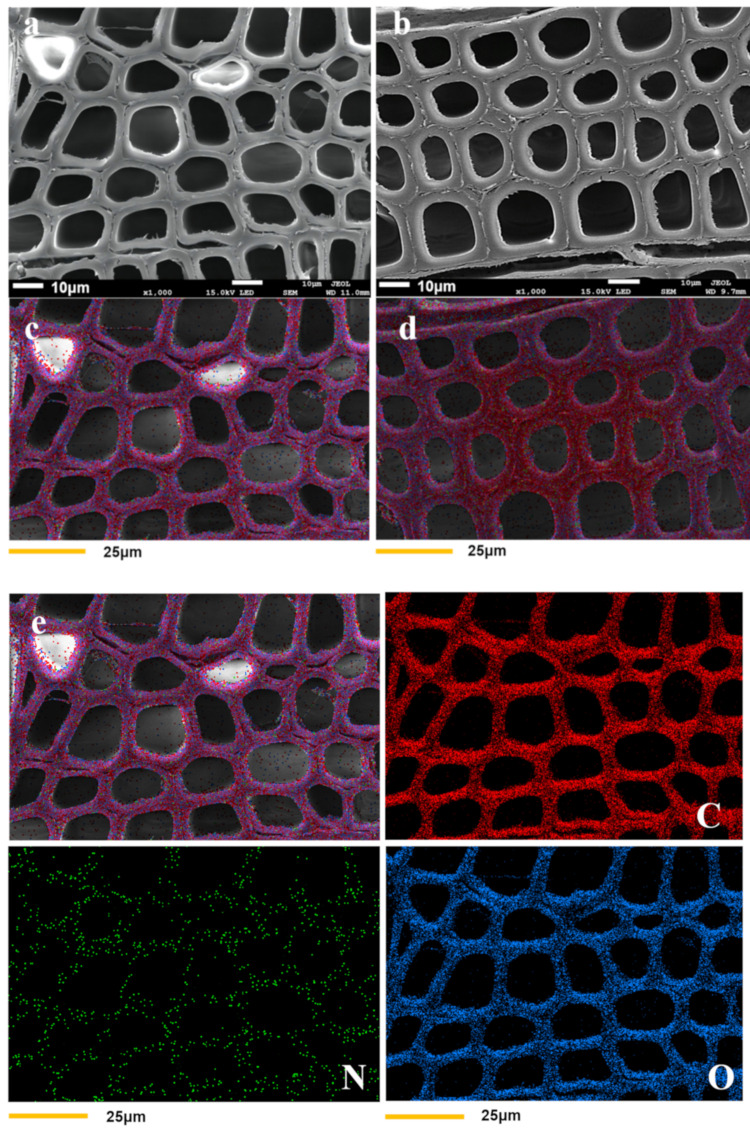
SEM morphologies of unmodified (**a**) and modified (**b**) wood samples, (**c**,**d**) are energy spectrum figures of untreated and treated wood, respectively. Samples (**e**,**f**) show the distribution of C, N, and O elements in the cell walls of unmodified and modified wood.

**Figure 3 polymers-14-03299-f003:**
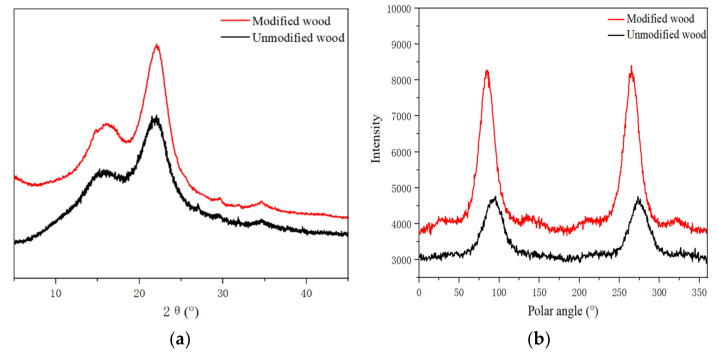
XRD patterns (**a**) and MFA (**b**) of modified and unmodified wood.

**Figure 4 polymers-14-03299-f004:**
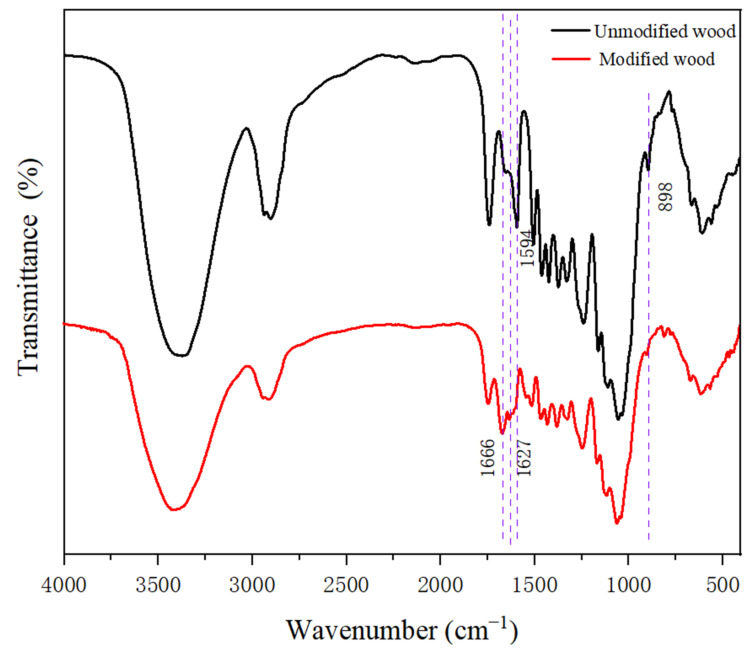
The FTIR spectra of the treated wood and untreated wood.

**Figure 5 polymers-14-03299-f005:**
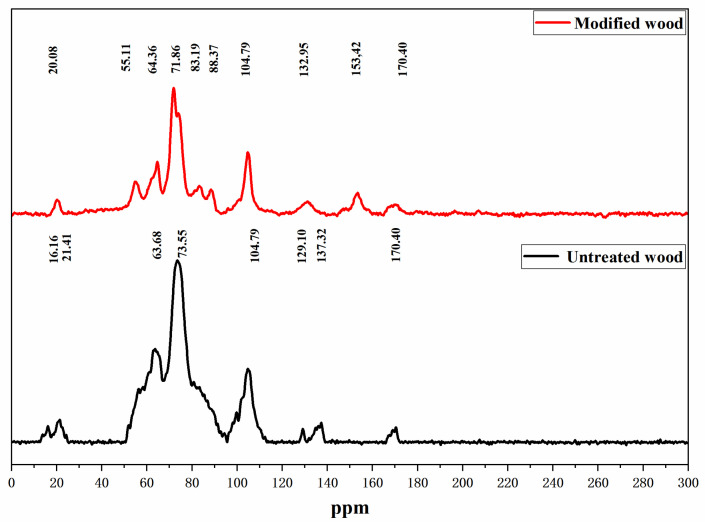
NMR spectra of untreated and treated wood.

**Figure 6 polymers-14-03299-f006:**
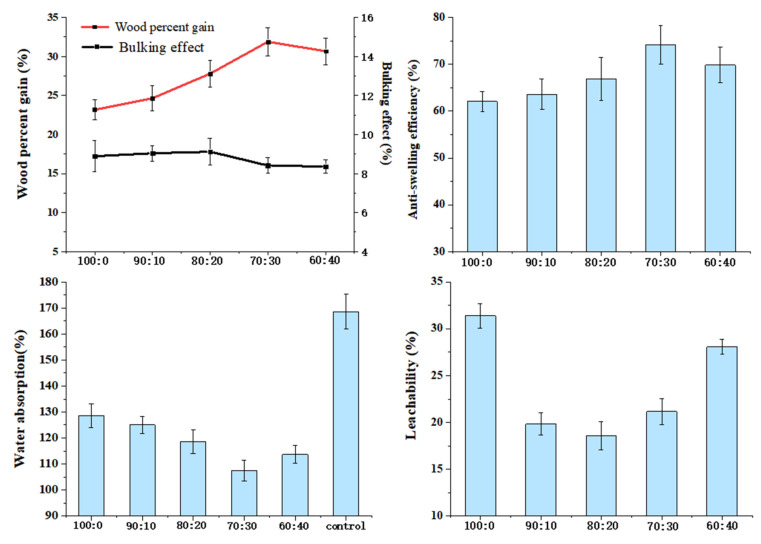
Effects of the ratio of HEMA and glyoxal on properties of wood.

**Figure 7 polymers-14-03299-f007:**
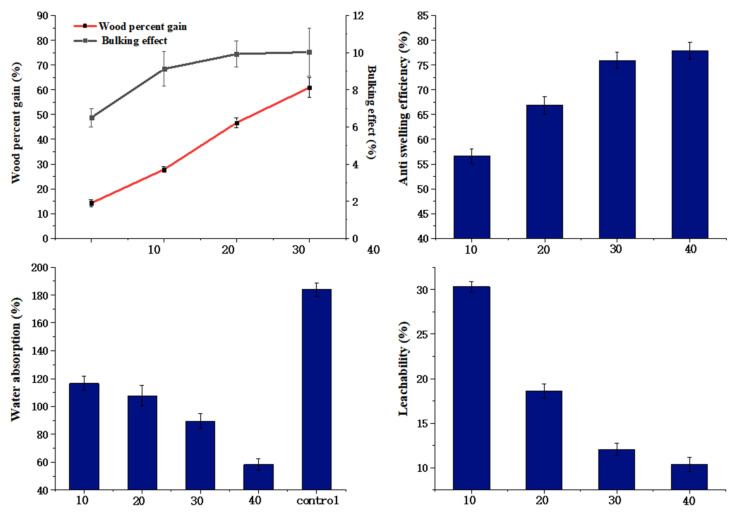
Effects of modifier concentrations on properties of wood.

**Figure 8 polymers-14-03299-f008:**
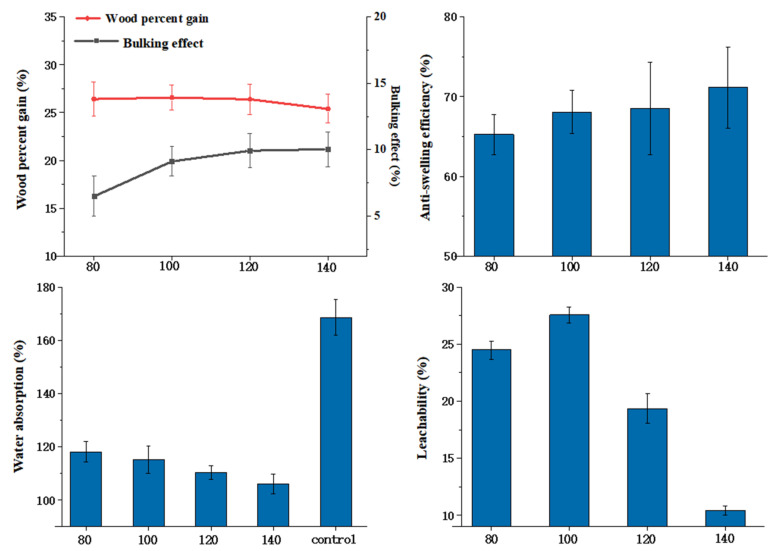
Effects of curing temperature on properties of wood.

**Table 1 polymers-14-03299-t001:** Parameters of treatments for each group of wood samples.

The Ratio of HEMA and GA	CaCl_2_/H_2_O_2_/(NH_4_)_2_S_2_O_2_ (%)	The Content of HEMA/GA (%)	Curing Temperature (°C)
100:0	3	10	80
90:10	3	20	100
80:20	3	30	120
70:30	3	40	140
60:40	3	-	-

**Table 2 polymers-14-03299-t002:** The data of untreated and treated wood cell wall thickness.

Labels	Total	Max./μm	Min./μm	Mean./μm
Untreated wood	120	5.34	4.38	5.04
Treated wood	120	8.40	7.20	7.74

**Table 3 polymers-14-03299-t003:** Parameters of treatments for each wood sample.

	Crystallinity	Microfibril Angle	Crystal Breadth
Untreated wood	43.15%	21.86°	1.74
Treated wood	54.77%	29.70°	2.49

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
