# Peer review of "Manufacturing and Characterization of Modified Wood with In Situ Polymerization and Cross-Linking of Water-Soluble Monomers on Wood Cell Walls"

_polymers, 2022, doi:10.3390/polym14163299_

Round 1
Author Response
Thank you for your compliments on our work. This manuscript has been extensively updated based on your suggestions.
Point 1: Page 2/15: please replace the verb research and skip the repetition
Response 1: Thanks. This paragraph has been rewritten, this sentence has been deleted in the revised version.
Point 2: Page 2/15: Materials and Methods- How many samples have been used per each property and final experiment?
Response 2: Thanks. Each property has 10 duplicate samples, and its statistical average value is taken. Information on sample quantity has been added to sections 2.2 and 2.4
Point 3: Page 5/15: Figure 3 and Table 3 are not mentioned in the text
Response 3: Thanks. A description of Table 3 and Figure 3 has been added to the XRD analysis. ”As can be seen from Table 3 and Figure 3b, the peak width of the X-ray diffraction pattern of poplar increased from 1.74 to 2.49, as well as the microfibril angle increased from 21.86° to 29.70°.”
Point 4: Page 10/15: In Fig.7 (not 9) an increase in modifier concentration results in a higher WPG
Response 4: Thanks. The description that should have been Figure 7 in the original manuscript was misexpressed as Figure 9, which has been revised to Figure 7
Point 5: Page 12/15: Figures 8 and 9 are not mentioned in the text
Response 5: Thanks. A description of Figures 8 and 9 has been added in sections 3.7 and 4. “As can be seen from the Figure 8, the curing temperature has no significant effect on BE.” and “A water-soluble monomer and crosslinker were injected into the wood cell walls and activated crosslinking reactions to form interpenetrating polymer network structures (Figure 9a). The polymer network blocked the partial pore and reduced hydroxyl of wood, increasing the transverse connection of wood simultaneously, increasing wood dimension stability significantly (Figure 9b). “
Point 6: Please insert the reference [41] in the manuscript
Response 6: Thanks. Reference 41 has been substituted for reference 14 in the original manuscript.

Reviewer 2 Report
Dear Authors,
I have read your manuscript and I think this work was done carefully and now the presentation of it can be improved. Please find comments and suggestions in the attached pdf file.
All best

Author Response
We appreciate your comments. We have corrected each of your comments on formatting, typos, images, etc. And we have answered each question individually.

Reviewer 3 Report
Manufacturing and characterization of modified wood with in situ polymerization and crosslinking of water-soluble monomers on cell walls
The possibility of using fast-growing species is often limited by their not entirely favorable physical and mechanical properties. On the other hand, fast-growing species make it possible to obtain a large amount of raw material in a short time. In this context, research on improving the properties of fast-growing species should be considered interesting and up-to-date. The use of 2-hydroxyethyl methacrylate and glyoxal for this purpose seems to be an interesting and new proposal compared to the currently used standard methods of chemical wood modification. Research in this area still carries a great cognitive and utilitarian potential. The introduction and the literature review are substantive and well introduce the reader to the subject of the article. The cited literature is up-to-date, mostly from the last few years. The work methodology is generally well described and does not raise any objections. Nevertheless, the authors should better describe the process of wood modification:
· When describing the modification process, they mention the initiators (CaCl2, H2O2 and (NH4)2S2O4). According to Table 1, their addition amounts to 3%, but there is no information as to whether they are applied simultaneously or otherwise.
· Shouldn't the last initiator be (NH4)2S2O3.
· In addition, initiators were not listed in the description of materials (point 2.1).
· The authors report that after modification, regardless of the variant, the samples were dried to constant weight at 103 ± 2°C. Is this treatment necessary in the case of modification of samples at temperatures of 120 and 140°C.
The article also lacks information on how many samples were modified and how many repetitions of individual tests were made. The research results are generally presented in a legible and understandable way. Nevertheless, the authors did not avoid certain inaccuracies:
· The part of Figure 2 on page 6 needs a better explanation. How does this drawing illustrate the distribution of C, N, and O elements in the cell walls?
· Figure 3 is not quoted in the text.
· In Figures 6, 7 and 8, standard deviations for individual research variants should be taken into account. This would facilitate the analysis of the research results presented by the authors.
· In the text, the Authors should correct the order of the figures - figure 9 is quoted first, and figure 8 later.
· Figure 9 should be included in the text, not after the conclusions.
The conclusions are adequate to the research carried out, however, they could be more detailed.
Taking into account the above remarks, the article requires a minor revision.
Author Response

(The authors gave the same response as above.)
